# Retrospective Study of Nosocomial Infections in the Orthopaedic and Rehabilitation Clinic of the Medical University of Lublin in the Years 2018–2020

**DOI:** 10.3390/jcm10143179

**Published:** 2021-07-19

**Authors:** Agnieszka Tomczyk-Warunek, Tomasz Blicharski, Rudolf Blicharski, Ryszard Pluta, Piotr Dobrowolski, Siemowit Muszyński, Ewa Tomaszewska, Mirosław Jabłoński

**Affiliations:** 1Chair and Department of Rehabilitation and Orthopaedics, Medical University of Lublin, Jaczewskiego (SPSK Nr 4) St. 8, 20-090 Lublin, Poland; a.tomczykwarunek@gmail.com (A.T.-W.); blicharskirudolf@gmail.com (R.B.); mbjablonski@gmail.com (M.J.); 2Laboratory of Ischemic and Neurodegenerative Brain Research, Mossakowski Medical Research Institute, Polish Academy of Sciences, Adolfa Pawińskiego 5, 02-106 Warszawa, Poland; pluta@imdik.pan.pl; 3Department of Functional Anatomy and Cytobiology, Faculty of Biology and Biotechnology, Maria Curie-Sklodowska University, Akademicka St. 19, 20-033 Lublin, Poland; piotr.dobrowolski@umcs.lublin.pl; 4Department of Biophysics, Faculty of Environmental Biology, University of Life Sciences in Lublin, Akademicka St. 13, 20-950 Lublin, Poland; siemowit.muszynski@up.lublin.pl; 5Department of Animal Physiology, Faculty of Veterinary Medicine, University of Life Sciences in Lublin, Akademicka St. 12, 20-950 Lublin, Poland; ewaRST@interia.pl

**Keywords:** nosocomial infections, hospital, healthcare-associated infections, rehabilitation, prevention, therapy control

## Abstract

Nosocomial infections appear in patients treated in hospital, which are not the initial cause of admission. A retrospective study concerning nosocomial infections was conducted to provide data about the amount, frequency and types of nosocomial infections in the rehabilitation ward in the Orthopaedics and Rehabilitation Clinic of Lublin. The study was conducted on a group of 49 patients that were admitted or transferred to the ward over a period of 20 months in the years 2018–2020. The patients and therefore the infections were divided by age, sex, time of hospitalization and the underlying disease. The study also provided data about the most frequent infection types in these patients, as well as the most commonly used drugs to treat those infections. The results showed that in fact all of the examined factors have an impact on the frequency of nosocomial infections appearance rates. Furthermore, results in the study showed that factors examined by the study also have an impact on what type of infection was present in these groups of patients.

## 1. Introduction

Nosocomial infections (NI) tend to appear in patients hospitalized in public hospitals, and are not the reason the patient was admitted in the first place [1]. NI may appear during the patients stay in the hospital, or even after the patient have been released from the hospital [2]. Other classifications that define NI are: infections occurring 48 h after the beginning of hospitalization, 3 days after patients release or 1 month after surgery [1].

NI are caused by bacteria, fungi, and viruses [2]. Frequently they are caused by bacteria, existing in the hospital as well as in the natural microbiome of a human organism. These bacteria cause infections especially at the moment of decreased immunity in patients [2]. The most common infections are blood, urinary tract, areas of surgery or pneumonia [3].

Data show, that in countries with high standard of healthcare, 7 out of 100 patients are diagnosed with NI, but in countries with lower standards of healthcare, the number of infected patients is higher, reaching up to 10 per 100 patients [2,4]. Clinical observations in the USA in 2003 presented that about two million patients annually admitted to hospital are diagnosed with NI [5]. Other studies have found that healthcare-associated infections are one of the top ten causes of patient death in the United States [6]. Subsequent studies on the epidemiology of HAI (hospital-acquired infection) showed that in developed countries, healthcare-associated infections occur in 5% to 15% of patients treated in hospitals, of which 9% to 37% occur in Intensive Care Units (ICU) [6,7,8]. In Europe, 4.6–9.3% of hospitalized patients are diagnosed with HAI [6].

NI are a major problem, because they are caused frequently by bacteria resistant to antibiotics, and increase recovery time for about 6.4 days to 12.7 days, while at the same time increasing the cost of hospitalization. NI also increases the risk of patient death [3,5]. In the USA alone, in Intensive Care Units about 90 thousand patients die because of infections, indicating that HAI are the most common cause of death in those units, not severe accidents [5].

HAI appear not only on the ICU, transplantology and surgery wards, but also on wards where surgical procedures are not conducted [9]. Development of NI is positively corelated with extended periods of hospitalization. Most common wards with prolonged patient persistence are rehabilitation wards, where patients are admitted with a variety of injuries or disabilities, which further translates to longer times of hospitalization [10].

Studies concerning the frequency of appearance of HAI are often conducted on the ICU, as well as on transplantology wards and infant care wards [2,4]. There is an abundance of studies showing the true number of HAI on rehabilitation wards, where patients with various injuries or after prior stay in the ICU are hospitalized, and their time of hospitalization is a big risk factor for the appearance of these infections [2,3].

Studies conducted in France in the 1990s showed a frequent appearance of HAI (most common in the urinary tract and the area of surgery) in patients on the rehabilitation wards and with long-term care [3]. The maximal general level of infections in this research reached 6.7% [3,11]. However, studies conducted in rehabilitation wards in the 2000s showed a general increase in the levels of the infections, some reaching from 8.2% to 12%, mostly concerning the urinary tract [6,12]. In 2005, 5.2% of patients had urinary tract infections [13].

Because of the few data and incomplete information, it seems crucial to gather data about up-to-date NI frequency in the rehabilitation ward in Poland. The available literature contains data from the 1990s, and the years 2000 and 2005, generally from France. It is crucial to remember that the prevention methods and the treatment itself changed dramatically in the last decade alone. The first step to eliminate the problem and subsequently decrease additional patients’ hospitalization costs is to present the severity of the issue.

The aim of the study is to evaluate the frequency of the appearance of HAI during 20 months between 2018–2020 on the rehabilitation ward. Additionally, it is to determine the specific groups of patients most prone to those infections and collect information about the most frequently used treatments on the rehabilitation ward. 

## 2. Materials and Methods

### 2.1. Place of Study

Retrospective study was performed in the Orthopaedics and Rehabilitation Clinic of Medical University of Lublin, Poland. Cumulatively in the years above, 1310 patients were hospitalized in this ward, from which about 290 were admitted for systemic rehabilitation, about 260 patients were admitted for neurologic rehabilitation and about 760 patients were admitted for surgeries and general diagnostics. Other types of patients hospitalized on this ward were patients admitted from previous hospitalization on other wards, patients transferred from the Hospital Emergency Department (HED), or patients with scheduled hospitalization involving systemic rehabilitation, neurologic rehabilitation and orthopaedic-trauma surgery.

The Orthopaedics and Rehabilitation Clinic contains 47 specialistic beds, which are located on one level of the hospital. The clinics are divided into two wings. Each wing contains its individual nursing point, a postoperative room with nurse supervision, and 7 three-person rooms. On one of the wings there are also 3 isolated single-person rooms, and the second wing has two isolated rooms and a room for occupational therapy.

### 2.2. Conducting of the Study

The retrospective study concerned patients admitted to the Orthopaedics and Rehabilitation Clinic, and was conducted over the course of 20 months from 01 July 2018 to 28 February 2020. The study was discontinued because of the SARS-CoV-2 pandemic, which almost completely stopped admissions. 

The study also focused on patients admitted from the admission room, HED, as well as patients transferred internally from other clinics. Together, the study covered 2474 patients. 

Information about the occurrence of the HAI was gathered basing on annual and semi-annual patient records on the registration of nosocomial infections carried out for the period covered by the study. The reports gathered the patients in a group, in which the infection was diagnosed by microbiological testing. The pathogens involved in the appearance of HAI are as follows:-*Staphylococcus aureus*-*Clostridioides difficile toxin A* and *B*-*Escherichia coli*-*Acinetobacter baumannii complex*-*Klebsiella pneumoniae* ssp. *pneumoniae*-*Enterococcus faecalis*-*Pseudomonas aeruginosa *-*Providencia stuartii *-*Staphylococcus epidermidis*-*Proteus mirabilis*-*Enterobacter cloacase* complex

From the reports, the following information were also gathered:-Patients age and sex;-Underlying disease;-Time of hospitalization;-Risk factors that might affect the appearance of HAI;-Types of microorganisms causing the HAI and their frequency;-Symptoms in infected patients;-Most common drugs used in treatment of these infections in the Orthopaedics and Rehabilitation Clinic.

### 2.3. Statistical Analysis 

Statistical analyses were performed using Statistica software (version 13.3, StatSoft, Poland). Data expressed on a qualitative scale were presented as the number and percentage of the sample. The chi-squared test (χ2) was used to compare the relationships between variables. Results were considered statistically significant when *p* < 0.05.

## 3. Results 

### 3.1. Basic Information

During the 20-month-period in which the study was conducted at the Orthopaedics and Rehabilitation Clinic, a total of 2474 patients were hospitalized. The patients were admitted both from the HED and the planned emergency room, as well as from other wards. During hospitalization in this period, 49 patients were diagnosed with HAI. Six patients were admitted from HED, 23 from the planned emergency room and 20 from other departments. In this group of patients, 72 different nosocomial infections and a total of 82 pathogens were detected. During hospitalization, 10 patients were diagnosed with more than one healthcare-associated infection. The most common type of nosocomial infection found in the study group were endogenous infections, the number of which was 64, and the remaining 8 were exogenous infections. The number of 69 healthcare-associated infections was detected more than 72 h after intake, and three infections were found before or up to 72 h after intake.

### 3.2. Sex

According to reports, nosocomial infections were detected in 17 women (34.69%) and 32 men (65.31%) (Figure 1). Women were diagnosed with 18 (25.00%) healthcare associated infections and 54 (75.00%) men (Figure 2).

### 3.3. Age

The age of patients who developed nosocomial infections ranged from 14 to 93 years. The mean age of hospitalized patients was 54 years. However, taking into account the division into sex, the age range for women was 46–93, with the mean age of 67 years. In the case of men, the age range was 14–86 years, and the mean was 48 years. In the case of the division of patients into age groups, in the first large group, nosocomial infections were detected in seven (14.29%) patients, while in the remaining groups, i.e., 30–49 years, 50–69 years and over 70 years, respectively in eleven (22.45%), twenty (40.82%), and eleven (22.45) NI (Figure 1).

When dividing the number of infections into individual age groups, in the first age group up to 30 years, a microbiological examination documented nine cases (12.50%) of healthcare-associated infections. In the age group between 30 and 49 years, 19 HAI (26.38%) were confirmed. The highest number of nosocomial infections, i.e., 28 cases (38.89%), was observed in the 50–69 age group. In the age group of 70 years and above, 16 cases (22.23%) of nosocomial infections were confirmed (Figure 2).

### 3.4. Time of Hospitalization

The mean hospitalization time of patients at the Orthopaedics and Rehabilitation Clinic with pathogens causing healthcare-associated infections was 76 days, varying from 6 to 262 days. Within 30 days of hospitalization, 14 (28.57%) patients were diagnosed with healthcare-associated infections, while for the remaining periods, i.e., 31–60, 61–90, 91–120, 121–150, 151–180 days and over 181 days nosocomial infections were diagnosed in nine (18.37%), nine (18.37%), seven (14.29%), four (8.16), three (6.12%), and three (6.12%) patients, respectively (Figure 1).

During 30 days of hospitalization, the patients were diagnosed with 15 nosocomial infections (20.83%). During the 31st to 60th day of hospitalization, the NI count was nine (12.50%), the same as for the period between 61 and 90 days. During 91 and 120 days, the number of nosocomial infections was eleven (15.28%). Eight (11.11%) healthcare-associated infections were detected over a period of 121 to 150 days. During the hospitalization for 151–180 days, seven NI (9.73%) were detected, while over 181 days, thirteen NI (18.05%) were confirmed (Figure 2).

### 3.5. Underlying Disease

The highest number of NI was noted in patients with paralysis, which was 28, representing 57.14% of all patients in this study. Infections of this type were found in fifteen patients (30.61%) with the tetraplegia, in seven (14.29%) hospitalized with the hemiplegia, and in six (12.24%) patients with the limb paralysis (Table 1).

However, in the case of other underlying diseases, the incidence of nosocomial infections was lower. In the case of primary bilateral gonarthrosis, other primary coxarthrosis, head injuries, idiopathic childhood scoliosis, spondylolisthesis, as well as unspecified complications of prostheses, implants and orthopaedic transplants, the NI was found in 12 patients (24.49%). This means that in this group, every second patient was hospitalized as a result of a diagnosis of the aforementioned disease entities (Table 1).

On the other hand, NI was diagnosed in one (2.04%) patient suffering from L5-S1 spinal disc inflammation due to tuberculosis, degenerative hip disease, discopathy, unclassified weight loss, primary bilateral coxatrosis, upper limb injury, fracture of the lumbar spine and femoral neck or femoral shaft. In total, this group included nine patients (18.51%) (Table 1).

Analysing our data, it can be seen that the number of infections is the highest in paralysis and amounts to 50, which constitutes 69.44% of all infections detected in the clinic. In tetraplegia, the incidence of nosocomial infections was 33 (45.83). In the case of paralysis of the lower limbs, the infection was detected in 10 (13.89%) health care centres. Seven (9.72%) healthcare associated infections were detected in hemiplegia. Four NI were diagnosed with unspecified complications related to prostheses, implants and orthopaedic transplants (Table 2).

Only two (2.78%), nosocomial infections were detected in: primary bilateral gonarthrosis, other primary coxarthrosis, head injuries, idiopathic childhood scoliosis and spondylolisthesis (Table 2).

However, only one (1.39%) nosocomial infection was detected in patients with: L5-S1 spinal disc inflammation due to tuberculosis, degenerative hip disease, discopathy, unclassified weight loss, primary bilateral coxatrosis, upper limb injury, fracture of the lumbar spine or femoral neck or femoral shaft (Table 2).

### 3.6. Infectious Factors

According to our retrospective investigations in the clinic during the 20 months, *Pseudomonas aeruginosa* caused hospital infections in the vast majority of instances, as this bacterium was found 16 times (22.22%). The second most frequently detected bacterium in patients was *Klebsiella pneumoniae* ssp. *pneumoniae*, identified in 15 patients (20.83%). Another NI bacteria that was documented 10 times (13.89%) in microbiological tests was *Proteus mirabilis*. *Clostridioides difficile toxin A and B* were diagnosed in nine cases in the group of hospitalized patients (12.50%). Microbiological tests revealed such bacteria as *Staphylococcus aureus*, *Escherichia coli*, and *Acinetobacter baumannii complex* seven times (9.72%). However, *Enterococcus faecalis* and *Enterobacter cloacase complex* were diagnosed four times (5.56%). *Providencia stuartii* was noted in two (2.78%) patients. In contrast, *Staphylococcus epidermidis* was diagnosed only once (1.39%) (Table 3).

### 3.7. Influence of Age, Hospitalization Time and Underlying Disease on the Number of Nosocomial Infections

Statistical analysis showed no significant influence of the patient’s age, hospitalization time and underlying disease. However, it should be noted that the obtained group of patients who were diagnosed with nosocomial infections in the 20-month period is relatively small, which may have influenced the lack of significant differences in the study (Table 4).

### 3.8. Risk Factors

Based on the presented data, it can be concluded that the case of urinary catheter was the risk factor causing the highest number of infections in the studied group of patients, as it was noted 61 times (84.45%). Another risk factor that triggered nosocomial infections in the study group was peripheral venous catheter, which was reported 59 times (81.94%). An additional factor that caused infection often was blood sampling, which was determined 47 times (65.28%). The peripheral puncture was marked 46 times (63.89%). The reports numerously indicated 20 (27.78%) previous infections and 13 (18.06%) surgical procedures. Implantation of a prosthesis marked seven (9.72%) reports. Previous surgeries and central venous puncture were marked six times. In five (6.94%) reports, a gastric tube as well as a respirator were indicated as risk factors. Multichannel catheters were marked in two reports (2.78%). In the case of renal failure, malnutrition, epidural catheter, blood transfusions and neoplastic disease, declining was observed after one (1.39%) report (Table 5).

### 3.9. Clinical Symptoms

Based on our data, it can be concluded that the most common symptom associated with nosocomial infections was fever, which was determined in 49 reports (69.06%). A frequent symptom was also exudation from the surgical site, which was reported 12 times (16.67%), and redness was noted in nine cases (12.50%). Diarrhoea was also noted in nine cases (12.50%), dysuria in six (8.33%), edema in four (5.56%), blood pressure decrease in two (2.78%) and nausea in one (1, 39%) (Table 6).

### 3.10. Clinical Form

The most common clinical form of pathology was urinary tract infections, which were observed in 31 patients (43.06%). There were also ten (13.89%) cases of gastrointestinal infections, as well as the same number of surgical area infections. In four cases (5.56%) urosepsis was noted. Pneumonia was also found in four (5.56%) cases. Three (4.17%) cases of bacteremia, sepsis and respiratory system infections were diagnosed. There were two (2.78%) incidences of skin and soft tissue infections, as well as one (1.39%) infection of the reproductive system and catheter vascular-related infection (Table 7).

### 3.11. Used Drugs

The drugs most frequently used in the treatment of nosocomial infections are antibiotics and they were accounted for 75.53% of all the therapy used. The most frequently administered antibiotic was meropenem, which was used in 13 (13.83%) cases, as well as metronidazole which was used 10 times (10.64%), the ciprofloxacin was used in 9 (9.57%) patients (Table 8). The remaining drugs are listed in Table 8.

## 4. Discussion

Nosocomial infections are important problems because they increase mortality and contribute to lengthening the hospitalization period of patients, thus increasing general healthcare costs [3,5]. The current literature is concerned with healthcare associated infections in intensive care unit, transplantology, burns and neonatal care units [2,4]. Research on nosocomial infections was also conducted in both rehabilitation centres and departments [3,9,13]. However, it should be noted that these studies are rare, only few attempts were made to analyse the problem [3,9,11,12,13]. The lack of current data on the frequency of nosocomial infections in rehabilitation wards is probably due to the fact that often the branches accepted only patients with current microbiological culture tests confirming the absence of such infections [9,13].

Our clinic uses a number of procedures aimed at preventing the development of nosocomial infections during the patient’s hospitalization. The preventive measures used in the clinic include training of medical personnel, maintaining hand hygiene and the use of protective gloves and gowns. Insulation is also used in high-risk patients, as well as additional masks and protective glasses. Whenever possible, single-use equipment is used to care for the patient. Medical devices are disinfected after each patient. The applied preventive measures are the standard procedures used in the world [14]. However, despite the application of appropriate preventive procedures against the development of HAI, nosocomial infections were detected in our clinic.

NI is most often caused by microorganisms belonging to the patients and that exist in the hospital [2]. In our work, 88.89% of the cases were endogenous infections caused by bacteria belonging to human microflora. Only in 11.11% of cases, nosocomial infections were caused by bacteria present in the hospital environment.

In our study, a higher frequency of nosocomial infections was observed in men than in women. Men represent 65.31% of the study group, while women 34.69%. Dividing the number of nosocomial infections according to sex, as many as 75% of all NI were diagnosed in men. This is confirmed by clinical studies which have demonstrated that males are more prone to nosocomial infections [15]. Research conducted in 2019 on a large group of patients by Aghdassi et al. [16] showed that men were also more sensitive to surgical site infections. The same studies showed a much higher rate of nosocomial infections after orthopaedic and trauma surgery [16]. The research conducted by Colbert et al. showed a lower frequency of pneumonia and inpatient sepsis in women with acute ischemic stroke [17]. It should also be noted that men are much more likely to suffer more severe accidents than women, which also significantly affects the decrease in immunity [18].

Age is another important risk factor in the development of nosocomial infections. This is confirmed by the research conducted by Bochicchio et al., which showed a significantly higher incidence of nosocomial infection in aging patients [19]. Other studies also confirm the significant influence of age on the emergence of NI in patients [19,20]. Different results showed a lower incidence of NI in patients over 60 years of age compared to other studies in this field [19,20]. Additionally, it has been shown that age significantly influences the appearance of NI in patients, but it is related to the presence of other risk factors, such as long hospitalization, chronic diseases, etc. [21].

In our study, patients were divided into four age groups up to 30, 30–49, 50–69 and over 70 years old. There was an increased frequency of nosocomial infections of 12.50%, 26.38%, 38.89% and 22.23%, respectively. The data show that the highest number of nosocomial infections was found in patients of 50–69 years. A lower number of infections was observed in patients over 70 years, i.e., 22.23%. However, this may be due to a significantly smaller group of this age patients and a more responsible approach of the staff in caring for this group, as they are more sensitive to complications. Based on the obtained results, it can be concluded that age generally influences the frequency of NI.

The duration of hospitalization increased the risk of nosocomial infections [10]. It exposes the patients to pathogens present in the hospital environment, which may be contaminated by another patient, hospital staff or by microorganisms that are infiltrating the hospital environment [22]. The appearance of NI in a patient as a result of a long stay in the hospital also causes the extension of hospitalization time. In another study, which was conducted in 68 hospitals, it was shown that this period will be extended by an average of 10.4 days [23].

In our study, the hospitalization time was divided into seven periods, i.e., up to 30, 31–60, 61–90, 91–120, 121–150, 151–180 and over 180 days, and the frequency of nosocomial infections was 20.83%, 12.50%, 12.50%, 15.28%, 11.11%, 9.73%, and 18.05%, respectively. The division of hospitalization time was standardized according to Liu et al. [24]. In the presented study, hospitalized patients were also divided in the same groups, and the following results were obtained: 28.57%, 18.37%, 18.37%, 14.29%, 8.16%, 6.12% and 6.12%. The results show that the number of infected patients decreases with the duration of hospitalization.

In the first period of hospitalization, the highest number of nosocomial infections can be observed, and in the next two periods they are decreased and maintained at the same level. It is most likely related to the detection time of NI and the introduction of appropriate treatment. The increase in the frequency of infections can also be observed in days 121–150, but in 151–180 days a decrease was noted. However, it can be seen that the highest number of infections was observed in the first stay, where the largest number of patients were hospitalized but in the period of hospitalization over 181 days, when fewer patients were hospitalized the number of infections was the highest, which suggests that the duration of hospitalization increases the risk of developing NI. In studies conducted by Sulzgruber et al., it can also be noted that the frequency of nosocomial infections associated with postoperative care increased significantly with the length of the patient’s pre-operative hospitalization [25].

The evaluation of the conducted studies shows that in patients with spinal cord injuries, who were also hospitalized in the orthopaedic and rehabilitation clinic, the risk of developing NI was higher [26,27]. These studies show that this type of infection occurs both during the first hospitalization and subsequent hospitalizations. This suggests that frequent and long duration stays in the hospital increases the risk of developing nosocomial infections in these patients [26,27].

Our study also shows that the highest number of infections was detected in patients with paralysis, which accounts for 57.14% of all patients included in the study, and the frequency was 69.44%. Due to the type of paralysis, the highest frequency of nosocomial infections was noted in patients with tetraplegia, at 45.83%. However, in the case of the limb paralysis or hemiplegia, the frequency of NI development was 13.89% and 9.72%, respectively. Observations may indicate a significant influence of motor limitation in patients on the emergence of NI development.

The presented study also shows that nosocomial infections occur not only in patients with damaged spinal cords, but also in other diseases, for example: complications after surgery, primary bilateral gonarthrosis, other primary coxarthrosis, head injuries, idiopathic childhood scoliosis, spondylolisthesis and other diseases listed in Table 2. It may be related to the weakness of the patient’s organism due to primary disease and the time of hospitalization [10].

In our work, *Pseudomonas aeruginosa* was a dominant infectious mediator. It was detected in 22.22% of hospitalized patients. Often the isolated bacterium was *Klebsiella pneumoniae*, which was detected in 20.83% cases. *Proteus mirabilis* was noted in 13.89% studies. *Clostridium difficile* was diagnosed in 12.50% cases. Other bacteria such as: *Staphylococcus aureus*, *Escherichia coli, Acinetobacter baumannii* complex, *Enterococcus faecalis*, *Providencia stuartii*, *Staphylococcus epidermidis*, and *Enterobacter cloacase* complex were also detected in patients hospitalized in our clinic.

According to Mylotte et al. the most frequently detected infections are caused by *Pseudomonas aeruginosa* and *Klebsiella pneumoniae* [9]. The study was conducted on a narrow group of patients, which compared NI in patients with spinal cord injury to patients without spinal cord injuries [9]. Our study covers a more diverse groups of patients, because the rehabilitation clinic treats not only patients with spinal cord injury, but also patients with other diseases that prolong hospitalization. Additionally, the Mylotte et al. study also presents that *Clostridium difficile* was responsible for nosocomial infections in 15% of patients hospitalized in acute rehabilitation unit [9]. In our studies, 12.50% of cases were also infected by *Clostridium difficile*.

However, the Girard et al. study presents the most frequent cases infected by *Staphylococcus aureus* and *Escherichia coli* [3]. Differences in diagnosed infectious bacteria between our study and the studies of Mylotte et al. and Girard et al. is probably caused by the different duration of the studies [3,9]. Our study lasted 20 months, and the study by Mylotte et al. was for 19 months, which gives a similar duration of study [6]. In contrast, the study by Girard et al. lasted from May 21 to June 15 of the same year, which could have contributed to the detection of other infectious bacteria in the studied group of patients [3].

The most common NI was urinary tract and infections of this type appear mainly in patients with spinal cord injury who have neurogenic bladders that require use of a urinary catheter [28]. According to data from 2014, 30% of all nosocomial infections are urinary tract infections connected with risk factors such as use of urinary catheter, surgery, other urological procedures, long hospitalization, age, male sex and chronic diseases [29].

Our patients also had 43.06% nosocomial urinary tract infection. Surgical areas infections and gastrointestinal tract infections were diagnosed for 13.89%.

The research conducted by Mylotte et al. [9] supports our data connected with the urinary tract, as well as skin and wound infections. In our study, it can be seen that gastrointestinal tract infections also appear frequently. In patients hospitalized in our clinic, which deals with the rehabilitation of individuals with various degrees of disability and undergoing orthopaedic and traumatic surgery, other clinical forms of hospital infections have also been diagnosed, such as: urosepsis, pneumonia, bacteremia, sepsis, respiratory system infections, skin and soft tissue infections, infections of the reproductive system and catheter vascular-related infections.

The most frequently identified risk factors in our work was the use of urinary catheters, which was the cause of 84.45% of all cases. Other frequent risk factors were peripheral venous catheter, blood sampling, peripheral puncture, and previous infections, which were noted with the following frequency: 81.94%, 65.28%, 63.89%, 40.28% and 27.78%, respectively.

The urinary catheter is always considered as the risk factor for the development of nosocomial infection, as it can introduce into the patient’s urinary tract microorganisms that contribute to the development of urinary tract infection [28]. In our study, the most common clinical form was urinary tract infection, which occurred as much as 81.94%, so urinary catheter infections are considered as a risk factor. Additionally, catheterization of the urinary bladder in men also increases the risk of developing NI, which is related to the anatomical structure of the urethra, which is much longer in men than in women. It additionally increases the risk of development nosocomial infections [29]. Studies indicate that urinary catheters in almost half of patients are incorrectly inserted and not replaced for long time [28,30]. The number of diagnosed pathogens responsible for causing HAI could also significantly affect the frequency of urinary tract infections. The most frequently exuded bacterium was *Pseudomonas aeruginosa*, which significantly increases the risk of developing urinary tract infections [31].

Peripheral venous catheters, peripheral puncture and blood sampling were also frequently considered as risk factors. Peripheral venous catheter and peripheral punctures are the main source of bacteremia and sepsis in patients during hospitalization [32].

Past infections are also marked as risk factors, as they significantly reduce the patient’s immunity and extend the duration of hospitalization, and thus contribute to an increased risk of developing further nosocomial infections [23].

Our observations also identified multichannel catheters, epidural catheters, central venous puncture, gastric tube, respirator, surgery, prosthesis implantation, previous operations, blood transfusion, kidney failure, malnutrition and cancer as risk factors consistent with the previous data [33,34,35].

In our examination the most frequent symptom of NI was fever, which was recorded in 69.06% of cases. A common symptom was also exudate from the surgical site, which was noted in twelve (16.67%) patients, and redness was noted in nine (12.50%) cases. Diarrhoea was also noted in nine (12.50%), dysuria in six (8.33%), edema in four (5.56%), blood pressure decrease in two (2.78%) and nausea in one (1, 39%) of the presented subjects.

In our work, we decided to include data about the drugs which were most frequently used to treat patients with diagnosed hospital infections. It can be concluded that the most frequently used therapeutic substances were antibiotics. Studies by other authors of similar topics support our use of antibiotics as the most common choice in the treatment of NI [36].

## 5. Conclusions

Based on our data, it can be concluded that nosocomial infections still occur in the Orthopaedic and Rehabilitation Clinic. Our research broadly describes the incidence of nosocomial infections in the clinic, and not only in terms of the infectious factors that cause them. Information on risk factors, clinical forms, symptoms and treatments were also presented. However, it should be noted that our study has limitations, as it was conducted on a small group of patients and the results were not statistically significant. However, the obtained data, shown in the form of figures and tables, presents the trends of the influence between different individual factors. On the other hand, it can be concluded that nosocomial infections are an important problem not only for intensive care units, but also for rehabilitation units and healthcare systems. It should be added that the risk associated with the development of nosocomial infections is frequently connected with the large number of drug-resistant bacteria. Our research is an introduction to a further study on a bigger group of patients. Our data suggest that nosocomial infections are a more frequent problem than previously thought. It seriously affects healthcare systems, and should trigger the development of new methods for discovery, prevention and treatments.

## Figures and Tables

**Figure 1 jcm-10-03179-f001:**
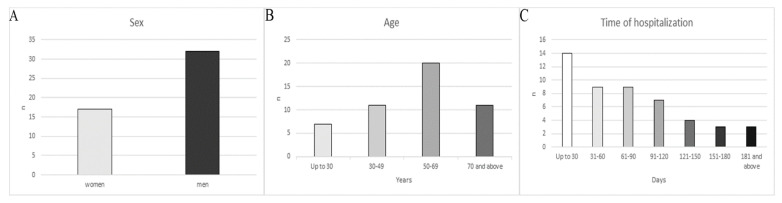
The number of patients diagnosed with nosocomial infections by sex (**A**), age (**B**) and time of hospitalization (**C**).

**Figure 2 jcm-10-03179-f002:**
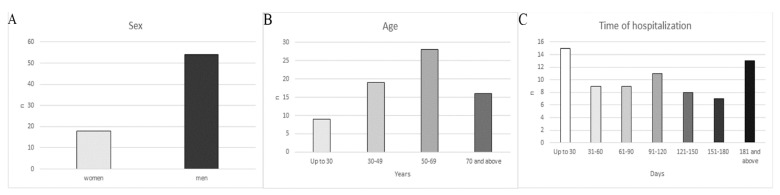
Classification of nosocomial infections by sex (**A**), age (**B**) and time of hospitalization (**C**).

**Table 1 jcm-10-03179-t001:** Basic diseases and the number of patients diagnosed with nosocomial infections.

Disease	N (%)
Hemiplegia	7 (14.29)
L5-S1 spinal disc inflammation due to tuberculosis	1 (2.04)
Degenerative hip disease,	1 (2.04)
Discopathy	1 (2.04)
Unclassified weight loss	1 (2.04)
Primary bilateral gonarthrosis	2 (4.08)
Other primary coxarthrosis	2 (4.08)
Primary bilateral coxatrosis	1 (2.04)
Upper limb injury	1 (2.04)
Head injuries	2 (4.08)
Fracture of the lumbar spine	1 (2.04)
Fracture of the femoral neck	1 (2.04)
Fracture of the femoral shaft	1 (2.04)
Unspecified complications of prostheses, implants and orthopaedic transplants	2 (4.08)
Idiopathic childhood scoliosis	2 (4.08)
Limb paralysis	6 (12.24)
Spondylolisthesis	2 (4.08)
Tetraplegia	15 (30.61)

Legend: N—number and % of patients with infections.

**Table 2 jcm-10-03179-t002:** Basic diseases and the number of detected nosocomial infections.

Disease	N (%)
Hemiplegia	7 (9.72)
L5-S1 spinal disc inflammation due to tuberculosis	1 (1.39)
Degenerative hip disease	1 (1.39)
Discopathy	1 (1.39)
Unclassified weight loss	1 (1.39)
Primary bilateral gonarthrosis	2 (2.78)
Other primary coxarthrosis	2 (2.78)
Primary bilateral coxatrosis	1 (1.39)
Upper limb injury	1 (1.39)
Head injuries	2 (2.78)
Fracture of the lumbar spine	1 (1.39)
Fracture of the femoral neck	1 (1.39)
Fracture of the femoral shaft	1 (1.39)
Unspecified complications of prostheses, implants and orthopaedic transplants	4 (5.56)
Idiopathic childhood scoliosis	2 (2.78)
Limb paralysis	10 (13.89)
Spondylolisthesis	2 (2.78)
Tetraplegia	33 (45.83)

Legend: N—number and % of nosocomial infections.

**Table 3 jcm-10-03179-t003:** The frequency of different bacteria causing nosocomial infections occurs.

Bacteria	N (%)
*Clostridioides difficile toksyna A i B*	9 (12.50)
*Staphylococcus aureus*	7 (9.72)
*Escherichia coli*	7 (9.72)
*Acinetobacter baumannii complex*	7 (9.72)
*Klebsiella pneumoniae* ssp. *pneumoniae*	15 (20.83)
*Enterococcus faecalis*	4 (5.56)
*Pseudomonas aeruginosa*	16 (22.22)
*Providencia stuartii*	2 (2.78)
*Staphylococcus epidermidis*	1 (1.39)
*Proteus mirabilis*	10 (13.89)
*Enterobacter cloacase complex*	4 (5.56)

Legend: N—number and % of nosocomial infections.

**Table 4 jcm-10-03179-t004:** Influence of age, time of hospitalization and diseases on the frequency of nosocomial infections.

	The Number of Infections = 1, N (%)	The Number of Infections = 2, N (%)	*p*
Age (mean ± SD)	54.40 ± 19.13	50.70 ± 16.57	0.563
Time of hospitalization (mean ± SD)	110.71 ± 80.91	106.80 ± 41.32	0.788
Underlying disease			
Hemiplegia	5 (8.06)	2 (20.00)	0.544
L5-S1 spinal disc inflammation due to tuberculosis	1 (1.61)	0 (0.00)	0.293
Degenerative hip disease,	1 (1.61)	0 (0.00)	0.293
Discopathy	1 (1.61)	0 (0.00)	0.293
Unclassified weight loss	1 (1.61)	0 (0.00)	0.293
Primary bilateral gonarthrosis	2 (3.23)	0 (0.00)	0.645
Other primary coxarthrosis	2 (3.23)	0 (0.00)	0.645
Primary bilateral coxatrosis	1 (1.61)	0 (0.00)	0.293
Upper limb injury	1 (1.61)	0 (0.00)	0.293
Head injuries	1 (1.61)	1 (10.00)	0.645
Fracture of the lumbar spine	1 (1.61)	0 (0.00)	0.293
Fracture of the femoral neck	1 (1.61)	0 (0.00)	0.293
Fracture of the femoral shaft	1 (1.61)	0 (0.00)	0.293
Unspecified complications of prostheses, implants and orthopaedic transplants	4 (6.45)	0 (0.00)	0.934
Idiopathic childhood scoliosis	2 (3.23)	0 (0.00)	0.645
Limb paralysis	9 (14.52)	1 (10.00)	0.913
Spondylolisthesis	2 (3.23)	0 (0.00)	0.645
Tetraplegia	28 (45.16)	5 (50.00)	0.955

Legend: N—number and % of nosocomial infections. Significance at *p* < 0.05 (The Chi-squared test).

**Table 5 jcm-10-03179-t005:** Influence of different risk factors on the development of nosocomial infections and their number.

Risk Factors	N (%)
Multichannel catheter	2 (2.78)
Peripheral venous catheter	59 (81.94)
Urinary catheter	61 (84.45)
Epidural catheter	1 (1.39)
Central venous puncture	6 (8.33)
Peripheral puncture	46 (63.89)
Gastric tube	5 (6.94)
Use of respirator	5 (6.94)
Surgical procedures	13 (18.06)
Implantation of a prosthesis	7 (9.72)
Previous surgeries	6 (8.33)
Blood transfusions	1 (1.39)
Blood sampling	47 (65.28)
Previous infections	20 (27.78)
Renal failure	1 (1.39)
Malnutrition	1 (1.39)
Neoplastic disease	1 (1.39)

Legend: N—number and % of nosocomial infections.

**Table 6 jcm-10-03179-t006:** The clinical symptoms in nosocomial infections.

Clinical Symptoms	N (%)
Diarrhoea	9 (12.50)
Nausea	1 (1.39)
Blood pressure	2 (2.78)
Fever	49 (69.06)
Dysuria	6 (8.33)
Redness	9 (12.50)
Exudation	12 (16.67)
Edema	4 (5.56)

Legend: N—number and % of nosocomial infections.

**Table 7 jcm-10-03179-t007:** Clinical form of nosocomial infections.

Clinical Form	N (%)
Bacteremia	3 (4.17)
Sepsis	3 (4.17)
Urosepsis	4 (5.56)
Surgical area infections	10 (13.89)
Skin and soft tissue infections	2 (2.78)
Gastrointestinal infections	10 (13.89)
Urinary tract infections	31 (43.06)
Respiratory system infections	3 (4.17)
Reproductive system infections	1 (1.39)
Catheter vascular-related infection	1 (1.39)
Pneumonia	4 (5.56)

Legend: N—number and % of nosocomial infections.

**Table 8 jcm-10-03179-t008:** Drugs used in the treatment of nosocomial infections.

Active Substance	Drug Class	N (%)
Amikacin	Antibiotic	8 (8.51)
Amoxicillin	Antibiotic	2 (2.13)
Ampicillin	Antibiotic	1 (1.06)
Cefotaxime	Antibiotic	1 (1.06)
Ceftazidime	Antibiotic	7 (7.45)
Ceftriakson	Beta-lactam antibiotic	4 (4.26)
Cefuroxime	Bacteriostatic	2 (2.13)
Ciprofloxacin	Antibiotic	9 (9.57)
Clindamycin	Antibiotic	3 (3.19)
Colistin	Antibiotic	2 (2.13)
Ertapenem	Antibiotic	2 (2.13)
Erythromycin	Antibiotic	1 (1.06)
Fluconazole	Antifungal	2 (2.13)
Fosfomycin	Antibiotic	4 (4.26)
Furaginum	Bacteriostatic	1 (1.06)
Gentamycin	Antibiotic	2 (2.13)
Herbal preparation	Diuretic drug	1 (1.06)
Levoflaxacin	Antibiotic	4 (4.26)
Linezolid	Antibiotic	1 (1.06)
Meropenem	Broad-spectrum antibiotic	13 (13.83)
Metronidazole	Antibiotic	10 (10.64)
Rifampicinum	Antibiotic	3 (3.19)
Sulfamethoxazole	Antibiotic	6 (6.38)
Vancomycin	Antibiotic	5 (5.32)

Legend: N—the number and % of drugs used.

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
