# Peer review of "Retrospective Study of Nosocomial Infections in the Orthopaedic and Rehabilitation Clinic of the Medical University of Lublin in the Years 2018–2020"

_jcm, 2021, doi:10.3390/jcm10143179_

Round 1
Reviewer 1 Report
The study aims to evaluate the frequency of appearance of the HAI during 20 months between 2018-2020 on the rehabilitation ward. Also, it is to determine the specific groups of patients most prone to those infections and collect information about the most frequently used treatments on the rehabilitation ward.
Based on this data, it can be concluded that nosocomial infections still occur in the Orthopaedic and Rehabilitation Clinic.
This study is well written, structured, and organized in a comprehensive and edifying way, treating the topic clearly and easily for the reader.
The tables and images are indispensable and well structured.
The only note from the reviewer is to revise the English grammar for some scattered errors that can be easily remedied.
I recommend the opportunity to deepen and cite the following reading that, on the contrary, documents how nosocomial infections by multi-resistant germs could be avoided, putting in place the shrewdness and the use of individual prevention devices considering the historical period in which we are.
PMID: 33498701
Author Response
I would like to thank you for your review. All your suggestions were important improve quality to our manuscript jcm-1255322. There were prevention methods in hospitalized patients added. Tis are the general rules and standards used around the world. Grammar errors were corrected.
Reviewer 2 Report
The most important references about epidemiology of NI, or better HAI , by WHO , CDC , ECDC is lacking.
Very interesting the focus about rehabilitation wards, HAI occurence, pathogens involved, risk factors.
The higher incidence of urinary infections, ( instead of pulmonary HAI as in the most recent literature) , it is related to the high number of patients with urinary catheter, as suggested by the authors . Pseudomonas Aeruginosa it was the first pathogen diagnosed for high percentage of urinary HAI and also suggesting a high circulation of nosocomial HAI pathogens ( Pseudomonas, Kl, Penumoniae, Acinetobacter spp, Cl ,Difficile , St. Aureus, ecc,). It will be important to know the % of MDRO's .
The difference in time of HAI occurence during the hospital stay , maybe more related to the very different groups ( in different wards ) of patients studied and of the underlying risk factors ( urinary catheters and other devices ) suggests further investigations between the different groups , also refferring to literature . About risk factors, it is very appropriate the analysis about different percentage of HAI in patients with devices: the results shown that maybe it will be important a deep analysis about preventive measures and specific bundle adopted by the health care workers
Preventive measures and specific references were never cited in the article
As suggested also by the authors further investigations will be made about compliance of antimicrobial stewardship programs with particular focus about second class of antibiotics and carbapenemes
Author Response
Thank you for your review. All your suggestions were included in the corrected manuscript. There was added epidemiology and preventive resources to prevent NI. In the manuscript jcm-1255322 interpretation of urinary tract infections and P. aeruginosa interpretation was added.
Suggestions according to deep analysis of the preventive methods used in the Orthopedics and Rehabilitation Clinic, the methods used by the health care system employees and %MDRO are valuable and will be used in the future articles. We will use homogenic group of patients in the future work. The aim of our recent work was to present patients with the various diseases and with different level of impairment. Another aim was to present the number of NI in such differential experimental group.